# Solar-Driven Interfacial Evaporation Using Bumpy Gold Nanoshell Films with Controlled Shell Thickness

**DOI:** 10.3390/ijms26136160

**Published:** 2025-06-26

**Authors:** Yoon-Hee Kim, Hye-Seong Cho, Kwanghee Yoo, Cho-Hee Yang, Sung-Kyu Lee, Homan Kang, Bong-Hyun Jun

**Affiliations:** 1Department of Bioscience and Biotechnology, Konkuk University, Seoul 05029, Republic of Korea; yoonhees@konkuk.ac.kr (Y.-H.K.); joh0302@konkuk.ac.kr (H.-S.C.); heu1997@konkuk.ac.kr (K.Y.); vltizk0052@konkuk.ac.kr (C.-H.Y.); sklee0504@konkuk.ac.kr (S.-K.L.); 2Gordon Center for Medical Imaging, Department of Radiology, Massachusetts General Hospital and Harvard Medical School, Boston, MA 02114, USA; hkang7@mgh.harvard.edu

**Keywords:** bumpy gold nanoshell, solar-driven interfacial evaporation, photothermal materials, absorption efficiency, evaporation rate

## Abstract

Metal nanostructure-assisted solar-driven interfacial evaporation systems have emerged as a promising solution to achieve sustainable water production. Herein, we fabricated photothermal films of a bumpy gold nanoshell with controlled shell thicknesses (11.7 nm and 16.6 nm) and gap structures to enhance their photothermal conversion efficiency. FDTD simulation of bumpy nanoshell modeling revealed that thinner nanoshells exhibited higher absorption efficiency across the visible–NIR spectrum. Photothermal films prepared by a three-phase self-assembly method exhibited superior photothermal conversion, with films using thinner nanoshells (11.7 nm) achieving higher surface temperatures and faster water evaporation under both laser and sunlight irradiation. Furthermore, evaporation performance was evaluated using different support layers. Films on PVDF membranes with optimized hydrophilicity and minimized heat convection achieved the highest evaporation rate of 1.067 kg m^−2^ h^−1^ under sunlight exposure (937.1 W/m^2^), outperforming cellulose and PTFE supports. This work highlights the critical role of nanostructure design and support layer engineering in enhancing photothermal conversion efficiency, offering a strategy for the development of efficient solar-driven desalination systems.

## 1. Introduction

Recently, the development of sustainable desalination technologies has been actively pursued to replace conventional desalination treatment facilities that incur enormous energy costs [1]. In particular, solar-driven interfacial evaporation, which utilizes solar energy, has attracted attention as a next-generation desalination solution because it offers efficient energy conversion while minimizing external energy input [2,3]. The main components of the solar-driven interfacial evaporation system are a photothermal material that absorbs light and converts it into heat, and a support layer that fixes the photothermal material and wicks water at the same time [4].

To date, various materials have been studied as photothermal materials, including carbon-based materials, semiconductors, and their hybrid composites [5,6,7,8,9]. Among them, metal nanostructures can strongly absorb light and efficiently convert it into heat energy through localized surface plasmon resonance (LSPR) [10]. In particular, gold nanostructures can maintain their performance even after long-term light irradiation due to their high stability as a photothermal material, and the absorption region can be adjusted by their morphological control. Based on these advantages, a solar-driven evaporation system using various types of gold nanostructures as photothermal materials has been reported [10,11].

Gold nanoshells are structures in which nanometer-scale gold shells are coated on a dielectric core and have been actively studied in various fields due to their unique plasmonic properties [12,13]. In particular, compared to individual gold nanoparticles, they exhibit a broad LSPR band due to hybridization between the plasmon resonance at the inner and outer surfaces of the nanoshell [14]. The broad LSPR band of gold nanoshells has been utilized for applications such as surface-enhanced Raman spectroscopy (SERS) substrates, photocatalysts, and laser desorption ionization matrices [15,16,17].

In our recent work, bumpy-shaped gold nanoshells on silica cores were studied, featuring controllable nanogap structures derived from the closely packed gold nanoparticles [18]. The thickness of the gold shell was adjusted to the tens-of-nanometers scale by controlling the gold precursor concentration during the growth process. The proposed bumpy gold nanoshell exhibited excellent performance as SERS substrates due to the electromagnetic hotspots generated at the nanogap structures. More recently, a high-throughput synthesis method for bumpy gold nanoshells was developed using a dual-channel syringe pump [19]. The proposed method can synthesize 10 mg per batch, which is 20 times higher than the conventional method. Furthermore, the shell thickness and gap structures can be similarly adjusted by controlling the volume of the gold precursor solution. Based on these previous works, bumpy gold nanoshells with adjustable morphology and optical properties are promising candidates for use as photothermal materials in the efficient fabrication of bulk-scale films.

Herein, we fabricated a photothermal film for solar-driven evaporation using bumpy gold nanoshells with controlled photothermal conversion efficiency. The morphology and corresponding photothermal activity of bumpy gold nanoshells were tuned through a seed-mediated growth process based on a previously reported high-throughput method. Using a three-phase self-assembly technique, the fabricated bumpy gold nanoshell layer was deposited onto a membrane to efficiently produce a photothermal film with highly concentrated nanoparticles. The growth conditions were optimized by simulating the absorption efficiency of nanostructure modeling and measuring the evaporation efficiency of the film under laser irradiation. This tendency was also observed under sunlight exposure, confirming the applicability of the system to solar-driven evaporation. Moreover, it was confirmed that heat convection could be minimized and the evaporation rate optimized by employing polymer membrane support with different hydrophilicities.

## 2. Results and Discussion

### 2.1. Bumpy Gold Nanoshells with Optimized Morphology for Photothermal Activity

A schematic illustration of the solar-driven evaporation system utilizing bumpy gold nanoshell films fabricated on a support layer as a photothermal film is shown in Figure 1a. Efficient interfacial evaporation is achieved by using films deposited on membranes with bumpy gold nanoshells with controlled shell thickness. Based on their excellent evaporation rate, vapor can be generated from seawater or polluted water, and fresh water can subsequently be obtained through a series of condensation processes.

The bumpy gold nanoshells used in this study were fabricated based on a previously reported high-throughput seed-mediated growth approach combined with an infusion system [19]. The shell thickness and gap structures of the bumpy gold nanoshells were adjusted by controlling the final concentration of the gold precursor (0, 1.0, 2.0 mM) and the reducing agent involved during the seed-mediated growth process, as reported in the literature. As shown in Figure 1b, as the concentration of the precursor increased, the size of the gold nanoparticles constituting bumpy gold nanoshells on the silica core (145.5 ± 11.7 nm) also increased, accompanied by close packing.

The shell thickness was calculated from the effective diameter of the 100 gold nanoparticles constituting each gold nanoshell. The average shell thicknesses were calculated to be 2.8 ± 0.5, 11.7 ± 2.3, and 16.6 ± 2.1 nm when the concentrations of gold precursor were 0, 1, and 2 mM, respectively (Figure 1c). Based on the core diameter, shell thickness, and the number of particles constituting the shell, as confirmed from the TEM images, modeling of each bumpy gold nanoshell was designed, and theoretical calculations were performed. To obtain the absorption efficiency (*η*_abs_), which is closely related to the photothermal conversion efficiency of the nanostructure, the absorption cross-section and scattering cross-section were calculated in each monitoring region under the Total-Field/Scattered-Field (TFSF) excitation source, and the extinction cross-section spectrum, which is the sum of the two, was obtained.

The experimental absorbance spectra of the bumpy gold nanoshell exhibited broad absorption across the visible–NIR region while the individual gold nanoparticles exhibited narrow absorption in the UV region (Appendix A). The calculated spectra for the two modeling, with particle diameters constituting the bumpy nanoshells of 11.7 and 16.6 nm, are shown in Figure 1d,e. The modeling of both regions exhibited strong bands in the near-infrared (NIR) region in the calculated scattering cross-section spectra with peaks at 934 nm and 836 nm, respectively. For the calculated absorption cross-section spectra, the modeling with a particle diameter of 11.7 nm showed two main bands at 708 nm and 934 nm, while the modeling with a particle diameter of 16.6 nm showed a main band at 647 nm and a weak shoulder around 828 nm. As shown in Figure 1f, when comparing the absorption efficiency, defined as the ratio of the absorption cross-section to the extinction cross-section at a specific wavelength, it can be confirmed that the modeling with a particle diameter of 11.7 nm shows higher absorption efficiency across the entire visible–NIR region. This trend in absorption efficiency, dependent on the particle diameter, is consistent with the reported simulation results for individual gold nanoparticles, i.e., smaller particles exhibit higher absorption efficiency [11]. The calculated absorption efficiencies support that bumpy gold nanoshells with thinner shell thicknesses exhibit higher photothermal conversion efficiency under identical light irradiation conditions, making them suitable materials for solar-driven evaporation systems.

### 2.2. Solar-Driven Evaporation Using Bumpy Gold Nanoshell Film

Based on the photothermal effect of bumpy gold nanoshells, Figure 2a characterizes the light source wavelengths that mainly contribute to water evaporation by measuring the time required to evaporate a water droplet under laser irradiation, monitored via a thermal camera. The photothermal conversion film utilized in this work was prepared using a modified three-phase self-assembly method based on a previously reported technique for fabricating individual gold nanoparticle monolayers. This method induced the formation of a monolayer through the self-assembly of bumpy gold nanoshells at the heptane–water–air interface, which was formed by injecting an aqueous particle solution into an octylamine solution (in heptane). Afterward, the monolayer formed by drying in a vacuum chamber was deposited onto a cellulose filter paper to prepare a photothermal conversion film, as shown in Figure 2b. In this way, a film with an area of 1825 mm^2^ was prepared from 1.5 mg of nanoparticles; that is, a film with an area of 12,170 mm^2^ can be prepared from one batch of bumpy gold nanoshells via the high-throughput synthesis method [20].

Two types of films fabricated with bumpy gold nanoshells with different shell thicknesses (t = 11.7 and t = 16.6 nm) were first irradiated with monochromatic lasers (635 nm and 980 nm) for 30 s, and then the temperature changes following the dropping of a water droplet (12 μL) were monitored using a thermal camera. Prior to the analysis, the reproducibility of the photothermal effect under the laser irradiation and sunlight by film fabrication was confirmed, as shown in Appendix A. For independently fabricated films, the reproducibility was evaluated as RSD ≤ 2.29% for laser irradiation and RSD ≤ 5.16% for sunlight irradiation. When a 635 nm laser was irradiated onto the bumpy gold nanoshell films for 30 s, the temperatures at the irradiated spots reached 89.4 °C (t = 11.7 nm) and 80.7 °C (t = 16.6 nm) for the two films, respectively. After a water droplet was added, the temperature decreased and reached equilibrium at an average of 38.4 °C (t = 11.7 nm) and 37.3 °C (t = 16.6 nm), respectively (Figure 2c). The maximum temperature, equilibrium temperature, and the temperature after water evaporation were all measured to be higher in the film of the bumpy gold nanoshell with the thickness of 11.7 nm, although the time required for the water droplet to evaporate and for the temperature to show a rapid change was the same (70 s) for both films. On the other hand, when the 980 nm laser was irradiated onto the bumpy gold nanoshell films for 30 s, the temperatures at the irradiated spots reached 160 °C (t = 11.7 nm) and 152 °C (t = 16.6 nm), significantly higher than those under 635 nm laser irradiation. Afterwards, the phase equilibrium temperatures were reached at an average of 54.4 °C (t = 11.7 nm) and 53.2 °C (t = 16.6 nm), respectively, immediately after the water droplet was added (Figure 2d). Similarly, the maximum temperature, equilibrium temperature, and the temperature after water evaporation were all measured to be higher for the film with the 11.7 nm bumpy gold nanoshells, while the time required for the water droplet to evaporate and show a rapid temperature change was the same (30 s) for both films. The difference in photothermal effect and evaporation times depending on the laser wavelength at the same power (0.1 mW/mm^2^) can be attributed to the superior NIR sensitivity photothermal effect of the thinner bumpy gold nanoshells. Based on these properties, bumpy gold nanoshells can be proposed as promising candidate materials for solar-driven evaporation systems, particularly those utilizing NIR light, which accounts for approximately 52% of the solar energy [21].

To apply bumpy gold nanoshells to a solar-driven evaporation system, the change in surface temperature of their films was monitored using a thermal camera under a practical solar irradiance environment (937.1 W/m^2^) (Figure 3a,b). As soon as the films were exposed to sunlight, the surface temperature rapidly increased, reaching 65% of the maximum temperature increment within 10 s. After 90 s of sunlight exposure, the surface temperatures increased to a maximum of 57.5 °C (*Δ*T~20.0 °C) and 53.9 °C (*Δ*T~21.1 °C) for films of bumpy gold nanoshells with shell thicknesses of 11.7 nm and 16.6 nm, respectively. This temperature increment was significantly higher compared to the maximum temperature of cellulose filter paper (34.3 °C, *Δ*T~2.7 °C) under the same sunlight exposure conditions, where only a slight increment was observed due to heat convection. To observe water droplet evaporation under sunlight, bumpy gold nanoshell films were first exposed to sunlight for 30 s, then a water droplet was added onto the film, and the temperature was monitored using a thermal camera (Figure 3c). The temperatures at both the film surface and at the spot where the water droplet was added were simultaneously monitored for films of bumpy gold nanoshells with shell thicknesses of 11.7 nm and 16.6 nm (Figure 3d).

Immediately after the addition of the water droplet, the temperature at the spot reached phase equilibrium at 30.1 °C (t = 11.7 nm) and 30.9 °C (t = 16.6 nm) for the bumpy gold nanoshell films, and subsequently converged to the temperature of the film surface as the water droplet evaporated. Based on the point where the temperature change per second (*Δ*T/s) rapidly increased to 0.2 or higher, it took 135 s (t = 11.7 nm) and 155 s (t = 16.6 nm), respectively, for the water to fully evaporate. These results demonstrate that the absorption efficiency of the materials constituting the film plays a key role in determining the evaporation rate under solar irradiation and suggest that bumpy gold nanoshell films can be effectively applied in practical solar-driven evaporation systems.

### 2.3. Evaporation Rate on Different Support Membranes

Based on the previously characterized photothermal effect, films of bumpy gold nanoshells with a thickness of 11.7 nm were fabricated on three types of support layers (cellulose, PTFE membrane, PVDF membrane) with different hydrophilicity for application in practical solar-driven evaporation systems. Figure 4a shows a schematic diagram of the film–water interface with different support layers. Films using cellulose filter paper as a support layer were easily wetted up to the surface of the bumpy gold nanoshell film due to water absorption, owing to the hydrophilic properties of the cellulose.

In this system, it is estimated that the heat generated by the photothermal effect is directly transferred to the water through heat convection, which is less advantageous than interfacial evaporation. Films utilizing a PTFE membrane as a support layer exhibit super hydrophobicity and show an extremely minimized contact area due to their strong repulsion against the water surface. This property is highly unfavorable for interfacial evaporation, even though the convention of heat generated by the film’s photothermal effect is minimized. Meanwhile, the PVDF membrane with intermediate hydrophilicity forms a stable interface when in contact with water, while the bumpy gold nanoshell film itself remains unwetted, thereby minimizing heat convection losses. To compare the evaporation rate for each support layer, a circular film with a diameter of 36 mm was exposed to sunlight (average 937.1 W/m^2^) for 2 h, and the evaporated water was measured by gravimetric analysis. The neat support layers, onto which no nanoparticles were deposited, showed evaporation rates of approximately 0.4 kg m^−2^ h^−1^ for all three types of supports, mainly driven by thermal convection. In contrast, compared to their neat counterparts, the films fabricated with bumpy gold nanoshells exhibited significantly enhanced evaporation rates: 0.909 kg m^−2^ h^−1^ (212%) for the cellulose filter paper, 0.574 kg m^−2^ h^−1^ (141%) for the PTFE membrane, and 1.067 kg m^−2^ h^−1^ (249%) for the PVDF membrane. The difference in evaporation rates depending on the support layers can be interpreted as reflecting heat convection and interfacial contact. This high evaporation rate of PVDF may be attributed to the porous structure of the membrane, which is maintained even after the formation of the particle film (Appendix A). In addition to the three types of membranes discussed in this study, various types of support, such as sponges, carbon materials, and polymer membranes, can also be considered. Moreover, the evaporation rate can be further enhanced by combining the system with a light-concentrating device such as a Fresnel lens.

## 3. Materials and Methods

### 3.1. Chemicals

Tetraethyl orthosilicate (TEOS), THPC, gold(III) chloride, polyvinylpyrrolidone (PVP, average molecular weight ≈ 10,000), 3-aminopropyltriethoxysilane (APTES), and ascorbic acid were purchased from Sigma-Aldrich (St. Louis, MO, USA). Ethanol (≥99.9%), heptane, and aqueous ammonia solution (25–28%) were purchased from Daejung Chemicals and Metals (Siheung, Republic of Korea). Deionized water was obtained using an AquaMAX Ultra 370 water purification system (Younglin Instruments, Anyang, Republic of Korea).

### 3.2. Characterization

The TEM sample was prepared by drop-casting the dispersion of s on the formvar-coated 200 mesh copper grids. TEM analysis was performed using a JEM-1010 instrument (JEOL, Tokyo, Japan) operated at an acceleration voltage of 80 kV. The UV-Vis absorption spectra were measured using a single-beam-type UV-Vis spectrophotometer (U-5100, HITACHI, Tokyo, Japan). The sizes of the gold nanoparticles were calculated from at least 100 particles in the TEM images using the ImageJ 1.52a software.

### 3.3. Synthesis of AuNP Seeds

The AuNP seeds were prepared using THPC as reducing agent and stabilizer [22]. First, the NaOH solution (0.2 M, 1.5 mL) was diluted by mixing with deionized water (47.5 mL) in a 100 mL round-bottom flask. Next, THPC (80%, 12 μL) and gold(III) chloride solution (50 mM, 1 mL) were added sequentially. As the ultra-small AuNP seeds are formed, the solution immediately turns dark brown upon addition of Au^3+^ precursor solution. The mixture was stirred vigorously for 1 h and then stored in a refrigerator at 4 °C.

### 3.4. Synthesis of Bumpy Gold Nanoshell

Bumpy gold nanoshells were prepared using a silica nanoparticle support. First, silica nanoparticles were prepared by the Stöber process [23]. Briefly, NH_3_ solution (4.5 mL) was mixed with ethanol (40 mL) in a round-bottom flask, and TEOS (1.6 mL) was added to initiate the sol-gel reaction. The reaction mixture was stirred vigorously for 1 h at 60 °C, followed by additional stirring for 4 h at room temperature. The resulting solution was washed several times with ethanol via centrifugation at 9000× *g*. Next, the resulting silica nanoparticles (12.5 mg) were modified with amino groups by mixing with APTES (15.5 μL) and NH_3_ solution (10 μL) in a vortex mixer. After stirring for 15 h, the mixture was washed several times with ethanol via centrifugation. The aminated silica nanoparticles (2 mg) were then mixed with the gold nanoparticle solution (9.8 mL), and the mixture was stirred overnight to prepare gold nanoparticle-seeded silica nanoparticles. After washing with deionized water, the resulting brown pellets were dispersed in ethanol. Using the as-prepared gold nanoparticle-seeded silica nanoparticles, bumpy gold nanoshells were synthesized using a high-throughput seed-mediated growth process optimized in our previous study [19]. The gold nanoparticle-seeded silica nanoparticles (10 mg) were dispersed in the PVP solution (100 mL, 1 mg/mL PVP) and transferred to a glass vial. The vial was then combined with an infusion pump system loaded with an aqueous solution of gold(III) chloride (50 mM) and ascorbic acid (100 mM). While the dispersion was being stirred, each solution was added simultaneously at a fixed flow rate (36.67 μL/min) for 1 h and 2 h to adjust the final volume of each solution to 2.2 mL and 4.4 mL. The resulting mixture was sequentially washed with deionized water and ethanol at 3000 g. The pellet was then re-dispersed in ethanol (10 mL) and stored in a refrigerator.

### 3.5. FDTD Simulation

The structural modeling and calculations were performed using the Ansys Lumerical FDTD software (1.9.3633). The FDTD method is based on the classical solution of Maxwell’s curl equations. The perfectly matched later (PML) boundary was utilized to calculate the absorption cross-section. Simulations were performed at a mesh size of 1 nm. Total-field/scattered-field (TFSF) was used as an excitation source with the wavelength range from 300 nm to 1500 nm. Structural modeling was undertaken as follows:(A)The surface of the silica sphere (diameter: 145.5 nm) is composed of 619 gold spheres with diameter: 11.7 nm;(B)The surface of the silica sphere (diameter: 145.5 nm) is composed of 308 gold spheres with diameter: 16.6 nm.

### 3.6. Fabrication of Photothermal Films with Bumpy Gold Nanoshell Layer

Photothermal films of bumpy gold nanoshells were formed through the reported water/oil/air three-phase self-assembly strategy [20]. A heptane solution of octylamine (10.4 μM, 4.5 mL) was added to a petri dish (Φ = 5.5 cm). Then, aqueous bumpy gold nanoshell dispersion (1 mg/mL, 1.5 mL) was gently injected into the bottom of the petri dish. Ethanol was added dropwise (total 1 mL) to the water/heptane interface. This process forms self-assembly islands of bumpy gold nanoshells at the water/air interface, resulting in a larger film. After removing the surrounding heptane, the film was dried in a vacuum chamber and deposited on a support layer (cellulose filter paper, PTFE membrane, and PVDF membrane).

### 3.7. Characterization of Photothermal Effect

The temperature changes were recorded with a portable-type IR thermal camera (Mini2, HIKMICRO, Hangzhou, China). Prior to temperature measurement, the self-calibration function built into the device was performed, and then distilled water at 25 °C was used as a reference. During the analysis of the photothermal effect, the temperature at the spot where the highest temperature occurred was recorded.

### 3.8. Evaporation of a Droplet of Water Under Laser and Sunlight Exposure

To investigate wavelength-dependent photothermal properties and interfacial evaporation, a bumpy gold nanoshell film was irradiated with 635 nm laser and 980 nm laser at the same power density (0.1 W/cm^2^). To compare the evaporation rate in a laser-irradiated environment, a water droplet (12 μL) was dropped on the irradiated spot for 30 s, and the temperature at that spot was monitored. The photothermal effect of the film in an actual sunlight exposure environment was investigated based on the maximum temperature spot using an IR thermal camera. To compare the evaporation rate, a water droplet (12 μL) was dropped on the center of the film after the 30 s exposure, and the minimum temperature at that spot was monitored.

### 3.9. Investigation of Evaporation Rate

To compare the evaporation rate dependent on the support layer in an actual solar irradiation environment, films (Φ = 36 mm) deposited with bumpy gold nanoshells (shell thickness = 11.7 nm) on three types of support layers (Cellulose filter paper, PTFE and PVDF membranes) were used. The films were floated upward on tap water (10 mL) in a petri dish to prevent heat convection. The system was exposed to sunlight (average 937.1 W/m^2^) for 2 h in a wind-shielded glass container under controlled temperature and humidity (T~20 °C, RH~60%). The mass of evaporated water was measured through gravimetric analysis, and the corresponding evaporation rate was calculated.

## 4. Conclusions

In summary, this study demonstrated that morphological control of the shell thickness and gap structure in bumpy gold nanoshells plays a significant role in determining their photothermal effect for the evaporation performance under solar irradiation. The thinner bumpy gold nanoshell exhibited superior absorption efficiency, resulting in strong photothermal effects, particularly under NIR excitation, which is suitable for utilizing sunlight. Films prepared from the bumpy gold nanoshell achieved efficient photothermal heating and high evaporation rates when exposed to the practical solar light, confirming their applicability in solar-driven desalination. Additionally, among the various support layers investigated, PVDF membranes were identified as the most effective substrate, due to their minimized heat convection and interfacial contact. Overall, the present study highlights the critical importance of morphological design and the corresponding absorption efficiency in optimizing the performance of solar-driven evaporation systems. The optimized bumpy gold nanoshell can be applied to various supports not addressed in this study, and there is room for further improvement in the evaporation rate.

## Figures and Tables

**Figure 1 ijms-26-06160-f001:**
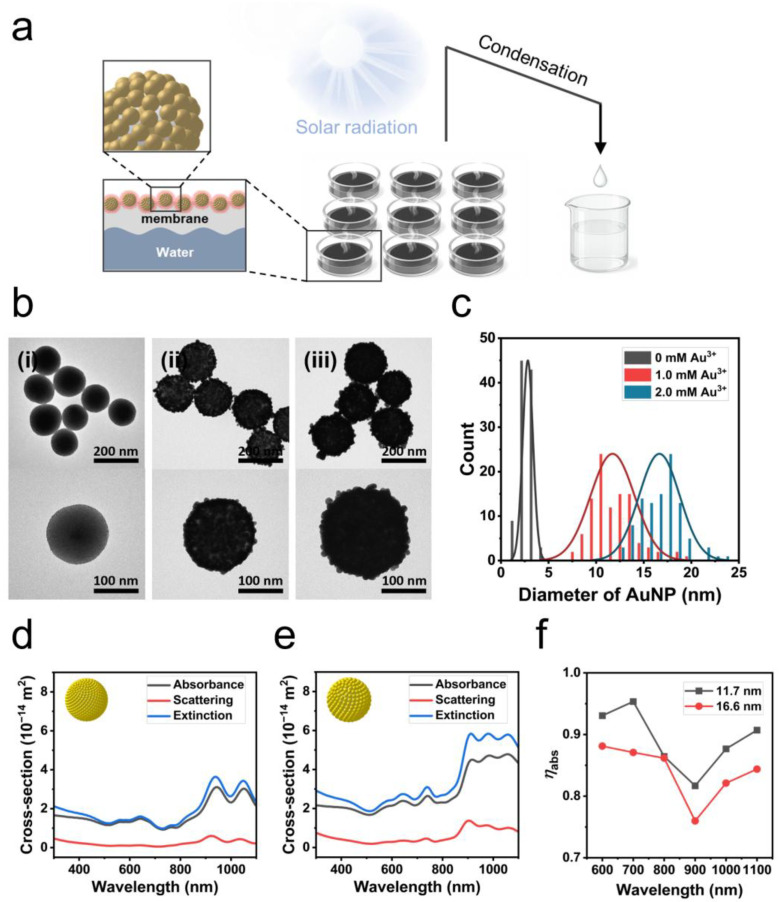
(**a**) Schematic illustration of solar-driven desalination system using photothermal membrane with bumpy gold nanoshell assembly. (**b**) TEM images of (i) gold nanoparticle-seeded silica nanoparticle and bumpy gold nanoshell grown with (ii) 1.0 mM and (iii) 2.0 mM Au^3+^ precursor. (**c**) Distribution of Au NP diameters corresponding to samples (i), (ii), and (iii) shown in (**b**) (*n* = 100). (**d**,**e**) Simulated spectra of absorption, scattering, and extinction cross-sections for bumpy gold nanoshell with different thicknesses (11.7 nm and 16.6 nm). (**f**) Absorption efficiency of bumpy gold nanoshells at different wavelengths.

**Figure 2 ijms-26-06160-f002:**
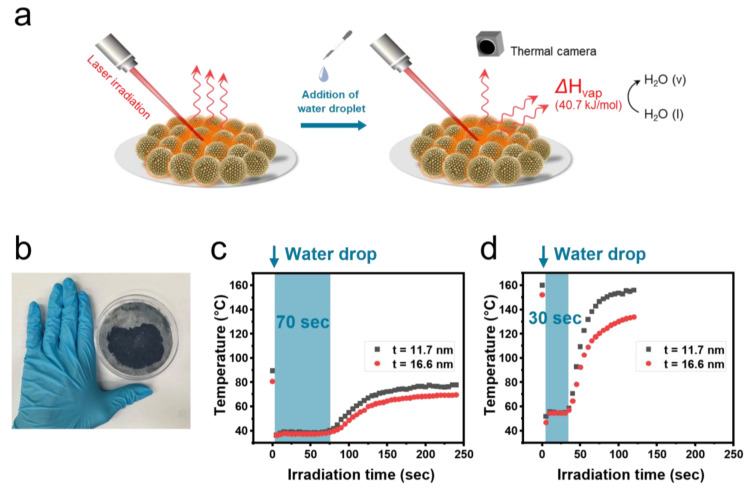
(**a**) Schematic illustration of laser-induced water evaporation on the surface of the bumpy gold nanoshell film. (**b**) Bumpy gold nanoshell film on cellulose filter paper prepared by three-phase self-assembly method. Temperature changes in a spot wetted with a water droplet under the irradiation of (**c**) 635 nm laser and (**d**) 980 nm laser on the film of bumpy gold nanoshell with different shell thicknesses (11.7 nm and 16.6 nm).

**Figure 3 ijms-26-06160-f003:**
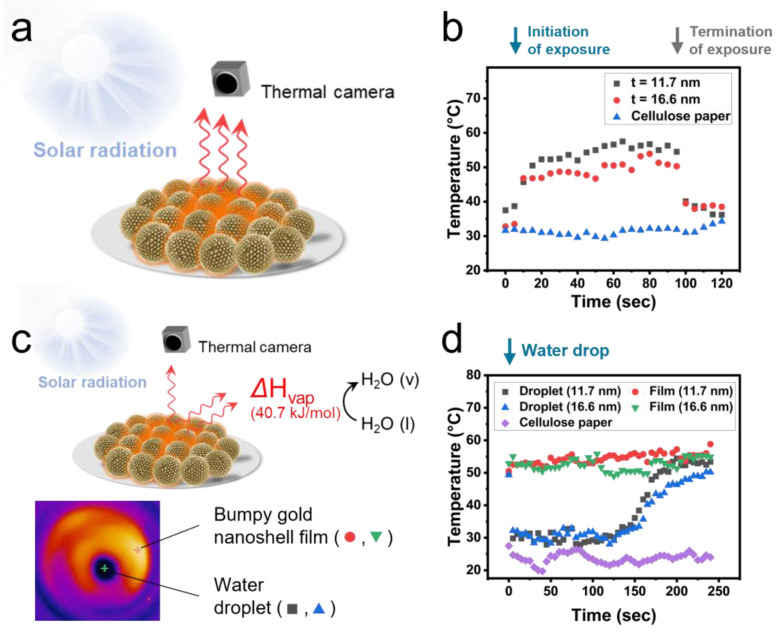
(**a**) Schematic illustration of heat generation of bumpy gold nanoshell film under sunlight exposure. (**b**) Temperature changes on filter paper (blue triangles), the film of bumpy gold nanoshell with shell thicknesses of 11.7 nm (black squares), and 16.6 nm (red circles). The films were exposed to sunlight for periods ranging from 5 to 95 s. (**c**) Schematic illustration of solar-driven evaporation of water on the film of bumpy gold nanoshell and thermal image captured immediately after the addition of a water droplet. (**d**) Temperature changes on sunlight-exposed filter paper (purple diamonds) and the film of bumpy gold nanoshells. The films were pre-exposed to sunlight for 30 s, and a droplet of water was added at the 5 s mark on the graph.

**Figure 4 ijms-26-06160-f004:**
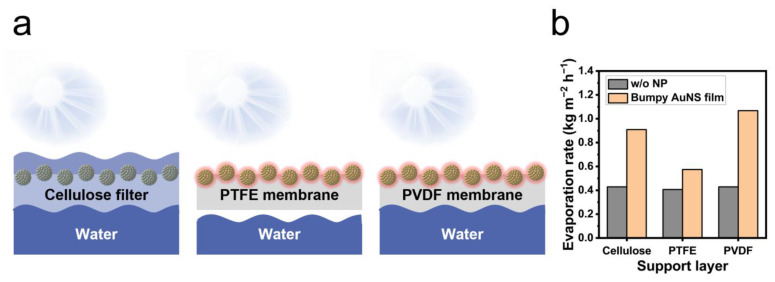
Comparison of the evaporation rate of the film of bumpy gold nanoshells with different support materials. (**a**) Schematic illustration of the cross-section of bumpy gold nanoshell films fabricated on different support materials (cellulose filter paper, PTFE membrane, and PVDF membrane) floating on water. (**b**) Percentage of the weight of evaporated water by bumpy gold nanoshell film with different support materials after 2 h of exposure to sunlight.

## Data Availability

The original contributions presented in this study are included in the article/Appendix A. Further inquiries can be directed to the corresponding author.

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
