# Peer review of "Solar-Driven Interfacial Evaporation Using Bumpy Gold Nanoshell Films with Controlled Shell Thickness"

_ijms, 2025, doi:10.3390/ijms26136160_

Round 1

Reviewer 1 Report

Comments and Suggestions for Authors

In their paper, Y-H Kim and co-authors aim to evaluate the influence of nanostructure design and support layer engineering in enhancing photothermal conversion efficiency for solar-driven desalination systems. They fabricated a photothermal film of bumpy gold nanoshells and simulated their absorption efficiency across the visible-NIR spectrum.

Although the work does provide some implications on how the shell thickness in bumpy gold nanoshells may affect their photothermal properties for the evaporation performance under solar irradiation, major revisions seem to be necessary to improve the soundness of the paper.

  1. Structure is a critical physical characteristic of the silica core/gold nanoshell particles. Given the broad size distribution of the silica core material and the gold nanoparticles, TEM measurements of 30 individual particles are statistically insufficient. There is no description in the methods section on how the shell thickness was measured/calculated and what the associated uncertainty is. Also, how was the “gap structures of the bumpy gold nanoshells” investigated?
  1. Since no evidence of close packing of the gold particles is provided, such assumption in the simulations is not justified. The use of average sizes for the core and shell further simplifies the model, making the simulation too idealistic. Experimental absorption measurements, therefore, should be an integral part of the study. Otherwise, the claim that a thinner shell exhibits higher photothermal conversion efficiency under identical light irradiation conditions may not be corroborated for the two as-synthetized bumpy gold nanoshells samples.
  1. The FDTD simulations may have erroneous/inconsistent assumptions. Silica sphere radius 68 nm (line 301) vs. silica core 145.5 nm (line 92). Gold sphere radius 11.9 nm (line 302) vs. particle diameter of 11.9 nm (line 116). The number of spheres assumed/calculated seems to be incorrect. Same comment for lines 303-304.
  1. The laser-induced and solar-driven water evaporation measurements were completed only on single samples. Given the variability of the constituent materials and the film preparation, replicate measurements would be highly necessary here to build confidence in the results.
  1. There is no description in the methods section of the temperature measurements/calibration.
  1. “The UV-Vis absorption spectra were measured using a single-beam-type UV-Vis spectrophotometer (U-5100, HITACHI, 260 Tokyo, Japan).” (lines 259-261). Such data does not seem to be presented in the paper.

In the opinion of the reviewer, the technical content and quality of the paper is, unfortunately, not suitable for publication in the International Journal of Molecular Sciences without major revisions.

Reviewer 2 Report

Comments and Suggestions for Authors

This is an interesting paper about the use of especially shaped nanomaterials for the capture of light energy for the evaporation of water I have some comments on the manuscript
1) The section in the introduction on nanomaterials as photothermal materials needs more references.
2) Please include characterization of the "seed" nanoparticles formed in the initial stages of the process. 
3) Please include some more characterization of the membranes used in the study. Are there any images showing the distribution of the gold nanoshells on the profile of the membranes?
4) Do you have any analysis of the durability of the nanoshells under heating? Have you measured the performance of the heating function of your membranes over multiple heating and cooling cycles?
5) This material is intended for desalinisation operations have you tested this material using real or simulated seawater? Has there been any study on the effect of seawater on your system? Please include information on that.  
6) The scale bar on the TEM images are not very clear at all. Please correct the images. 
